# A multifaceted interplay between virulence, drug resistance, and the phylogeographic landscape of *Mycobacterium tuberculosis*

Igor Mokrousov,[1] Tatiana Vinogradova,[1,2] Marine Dogonadze,[2] Natalia Zabolotnykh,[2] Anna Vyazovaya,[1] Maria Vitovskaya,[2] Natalia Solovieva,[2] Boris Ariel[2]

**ABSTRACT** Latin-American Mediterranean (LAM) family is one of the most significant and global genotypes of *Mycobacterium tuberculosis*. Here, we used the murine model to study the virulence and lethality of the genetically and epidemiologically distinct LAM strains. The pathobiological characteristics of the four LAM strains (three drug resistant and one drug susceptible) and the susceptible reference strain H37Rv were studied in the C57BL/6 mouse model. The whole-genome sequencing was performed using the HiSeq Illumina platform, followed by bioinformatics and phylogenetic analysis. The susceptible strain H37Rv showed the highest virulence. Drug-susceptible LAM strain (spoligotype SIT264) was more virulent than three multidrug-resistant (MDR) strains (SIT252, SIT254, and SIT266). All three MDR isolates were low lethal, while the susceptible isolate and H37Rv were moderately/highly lethal. Putting the genomic, phenotypic, and virulence features of the LAM strains/spoligotypes in the context of their dynamic phylogeography over 20 years reveals three types of relationships between virulence, resistance, and transmission. First, the most virulent and more lethal drug-susceptible SIT264 increased its circulation in parts of Russia. Second, moderately virulent and pre-XDR SIT266 was prevalent in Belarus and continues to be visible in North-West Russia. Third, the low virulent and MDR strain SIT252 previously considered as emerging has disappeared from the population. These findings suggest that strain virulence impacts the transmission, irrespective of drug resistance properties. The increasing circulation of susceptible but more virulent and lethal strains implies that personalized TB treatment should consider not only resistance but also the virulence of the infecting *M. tuberculosis* strains.

**IMPORTANCE** The study is multidisciplinary and investigates the epidemically/clinically important and global lineage of *Mycobacterium tuberculosis*, named Latin-American-Mediterranean (LAM), yet insufficiently studied with regard to its pathobiology. We studied different LAM strains (epidemic vs endemic and resistant vs susceptible) in the murine model and using whole-genome analysis. We also collected long-term, 20-year data on their prevalence in Eurasia. The findings are both expected and unexpected. (i) We observe that a drug-susceptible but highly virulent strain increased its prevalence. (ii) By contrast, the multidrug-resistant (MDR) but low-virulent, low-lethal strain (that we considered as emerging 15 years ago) has almost disappeared. (iii) Finally, an intermediate case is the MDR strain with moderate virulence that continues to circulate. We conclude that (i) the former and latter strains are the most hazardous and require close epidemiological monitoring, and (ii) personalized TB treatment should consider not only drug resistance but also the virulence of the infecting strains and development of anti-virulence drugs is warranted.

**KEYWORDS** *Mycobacterium tuberculosis*, virulence, multidrug resistance, murine model, phylogeography

Address correspondence to Igor Mokrousov, igormokrousov@yahoo.com.

The authors declare no conflict of interest.

See the funding table on p. 15.

*M*ycobacterium tuberculosis sensu stricto is an important human pathogen with a clonal, hierarchical, and heterogeneous population structure. Its evolutionary trajectory, genomic diversity, and phylogeography have been shaped by multiple factors, in particular, the coevolution with its human host and interaction with the human immune system. On the phylogenetically large scale, *M. tuberculosis* includes several lineages, while L1 to L4 are the most global and significant, although differ in their geographic patterns. In particular, Lineage 4 (Euro-American) is spread in many world regions, although remains low prevalent in most of Asia. Based on the available knowledge of phylogenetics, phylogeography, and human history, it was speculated that, most parsimoniously, Lineage 4 likely originated in the Eurasian heartland more than 3,000 years ago coinciding with the origin of the Indo-European languages (1). The most known families within Lineage 4 are LAM, Ural, Haarlem, S (all originated in Europe), L4.5 (endemic in South China), and diverse families endemic in Africa and named after different African countries where they likely originated during the colonial period.

On the global scale, Latin-American Mediterranean (LAM) family is perhaps the second most studied genotype of *M. tuberculosis* after the Beijing genotype. The LAM family was discovered more than 20 years ago in the seminal work of Sola et al. (2) based on the phylogenetic analysis of the large CRISPR-spoligotyping data set, and the name reflected the strains' origins. LAM is prevalent in the Americas, Europe, Russia, and some parts of Africa. Initially defined by spoligotyping, the LAM prototype spoligotype profile is SIT42 with deleted signals 21–24 and 33–36. It should be kept in mind that SIT42 presents a stable profile because it includes genetically distant isolates, that is, this SIT is found across all LAM phylogeny and thus does not correlate with any particular subfamily within LAM. With regard to the medically important features, it should be noted that different LAM strains were shown to be associated with multidrug resistance in different settings including Russia (3–6), North and South America (7, 8), and Africa (9).

Inside the LAM lineage, there are sublineages mainly defined by large genomic deletions (RD, regions of difference) and other markers (Fig. S1). LAM RD-Rio sublineage was first found in Rio-de-Janeiro, Brazil, and was shown to be MDR associated (7). The LAM population in Northern Eurasia is overwhelmingly (>90%) represented by sublineage RD115/LAM-RUS, that is, isolates with RD115 deletion and specific IS*6110* insertion in the *plcA* gene (10, 11). Several studies demonstrated the association of the Russian LAM strains with drug resistance (3, 5, 6). Drug-resistant spoligotypes SIT252 and SIT266 (both belonging to LAM-RUS) were previously termed as emerging resistant LAM clones with a potential risk of their wider spread over time hence the importance of epidemiological surveillance (12–14).

The spread of the particular genotypes is also impacted by external factors not related to the biological properties of the *M. tuberculosis* strains. LAM populations are similar in Mongolia and Eastern Siberia, Russia and are dominated by the LAM-RUS VNTR types. However, Russian but not Mongolian LAM strains are MDR associated which was attributed to insufficient health control and non-adequate TB treatment in Russia (6).

Unlike the Beijing genotype, the virulence of the LAM genotype has been little studied in animal and macrophage models to date. A Russian study by Zemskova et al. (15) demonstrated that Beijing, LAM, and reference H37Rv strains showed similar and high growth in the infected murine macrophages compared to Ural strains. The increased virulence of the Russian LAM strains similar to that of the Beijing strains in the *in vivo* mouse model was also demonstrated by the same group (16). It should be noted that, in spite of its certain limitations, the mouse model of tuberculosis infection is a classical one, and remains useful and meaningful (17–19). Recent studies of the particular Beijing subtypes demonstrated that hypervirulent and highly lethal (in murine model) strains may also lead to higher lethality in humans (20, 21).

In spite of well-known limitations, mice remain useful as model animals to study TB pathogenesis (22, 23). Here, we used the C57BL/6 mouse model and intravenous tail injection of the bacterial suspension according to the well-established and validated methodology previously used by us and others to study *M. tuberculosis* virulence

(20, 23–25). We studied genetically and epidemiologically distinct strains of the LAM family in Russia that represented highly drug-resistant spoligotypes SIT252 and SIT266, low-prevalent and susceptible SIT264, and resistant and geographically widespread SIT254. The available whole-genome sequencing (WGS) data were used to gain insight into genetic variation possibly underlying the observed pathogenetic patterns. Finally, the obtained results were interpreted in light of the dynamic phylogeography of the studied genotypes.

## MATERIALS AND METHODS

### *M. tuberculosis* strains

The strains were recovered from the respiratory material of patients with infiltrative or fibrous-cavernous pulmonary TB time and place of sampling. Based on the spoligotyping and genotyping of the cluster-specific markers, they were assigned to different spoligotypes of the LAM family and its LAM-RUS branch. Spoligotyping and detection of other molecular markers of the LAM and LAM-RUS were performed as described previously (4, 7, 10).

Drug susceptibility testing of the strains was carried out using the method of absolute concentrations (Order No. 109 of the Ministry of Health of the Russian Federation) and/or using the automated system BACTEC MGIT 960.

### Whole genome sequencing

The bacterial DNA was submitted to whole-genome sequencing performed using the Illumina HiSeq4000 platform. Whole-genome, paired-end sequencing on the HiSeq4000 platform was done using NEBNext Ultra, MiSeq Reagent v3, and PhiX Control v3 kits (Illumina). DNA libraries were prepared using ultrasound DNA fragmentation and NEBNext Ultra DNA Library Prep Kit for Illumina (New England Biolabs).

Data for the *M. tuberculosis* sequenced genomes were deposited in the NCBI Sequence Read Archive (project number PRJNA886055).

SAM-TB online tool (https://samtb.uni-medica.com/index) was used for SNP calling and phylogenetic analysis while SNPs in PE/PPE and resistance genes were excluded from phylogenetic analysis. The resulting concatenated FASTA file (678 variant nucleotide positions) was used to build a maximum likelihood tree using the MEGA7 package (https://www.megasoftware.net/), under default parameters and 1,000 bootstrap replicates.

Geneious 9.0 package (Biomatters Ltd) was additionally used for mapping the reads to the genome of reference strain H37Rv (NC_00962.3).

SAM-TB online tool was used for genotypic detection of drug resistance. MDR, pre-XDR, and XDR phenotypes were defined according to the updated World Health Organization definitions: MDR are strains resistant to isoniazid and rifampicin; pre-XDR —resistant to isoniazid, rifampicin, and fluoroquinolone; XDR—resistant to isoniazid, rifampicin, fluoroquinolone plus bedaquiline, and/or linezolid (26).

The significance of amino acid substitutions was assessed using Point Accepted Mutation 1 values calculated by PhyResSE online tool (https://bioinf.fz-borstel.de/mchips/phyresse/). The SIFT tool was used to predict whether an amino acid substitution affects protein function based on sequence homology and the physical properties of amino acids (https://sift.bii.a-star.edu.sg/index.html). As potentially biologically meaningful mutations, we defined those with significant SIFT *P* value or short indels. We also considered mutations in the putative promoter regions.

### *M. tuberculosis* strains used for *in vivo* experiment

The mouse model study included five *M. tuberculosis* strains—four clinical LAM isolates and a reference laboratory strain H37Rv. The reference strain H37Rv was received from the Collection of the Scientific Center for the Expertise of Medicinal Products, Moscow,

Russia (and initially received from the Institute of Hygiene and Epidemiology, Prague, Czech Republic).

The strains were cultured on Loewenstein-Jensen medium (Becton Dickinson, USA) and after 3 weeks of incubation at +35°C, the culture was suspended in a physiological solution with 15% glycerol, placed in cryovials, frozen, and stored at −80°C. Three weeks before the experiment, all strains were recultured on the Löwenstein-Jensen medium.

## Experimental animals

All experimental procedures were carried out following National guidelines (Rules for working with laboratory rodents and rabbits, 2016) and were reviewed and approved by the Ethical Committee of the St. Petersburg Research Institute of Phthisopulmonology (Protocol 48.1 of 20 May 2019). In total, 210 C57BL/6 male mice (weight 16–18 g) were used in all experiments. The mice were obtained from the Andreevka laboratory animal nursery (Moscow region, Russia).

The animals were kept under the conditions of a certified animal facility at the St. Petersburg Research Institute of Phthisiopulmonology using NexGen Mouse IVC Cage & Rack system with built-in ventilation and air conditioning system. Before the start of the study, laboratory animals were quarantined for 14 days. The body weight was monitored weekly using an Adventurer electronic balance (OHAUS Corporation, USA). The criteria for inclusion of the animals in the experiment were as follows: positive dynamics of the body weight of animals during the quarantine period and the absence of visible symptoms of the disease.

## Animal study design

In the animal model study, we followed the methodology that was previously described (20, 23). A mycobacterial suspension for infecting mice was prepared ex tempore from 3-week-old strains. The infecting dose was $10^6$ CFU in 0.2 mL of saline buffer per mouse. A suspension of mycobacteria was inoculated into the lateral tail vein of the animals.

The animals were divided into two series to study the virulence of strains and survival of animals infected with mycobacteria.

In the first series, 180 mice were observed, 36 mice for each of the five studied *M. tuberculosis* strains. The animals were euthanized in groups of six mice at days 7, 14, 21, 28, 56, and 112 post-infection (p.i.). Next, an autopsy and a sterile sampling of the lungs and spleen were performed for further culturing of mycobacteria. Lungs and spleen were weighed for calculation of the weight coefficients. The lungs and spleens of the mice were aseptically removed. The weight of each lung was calculated and the gross anatomic picture was taken to document the extent of the lesions.

The weight coefficients for lungs and spleen were calculated based on the ratio of organ weight and animal body weight and expressed in conventional units: *weight coefficient = (organ weight/animal body weight) × 100.*

The second series consisted of 100 mice, that is, 20 per strain. The natural death of animals was recorded after infection. The animals that died during the study were subjected to an autopsy with an examination of the internal organs. The experiment was ended when a lethality rate of 100% was reached in the group of mice infected with the reference strain *M. tuberculosis* H37Rv. In our experiment, the observation period was 196 days. The lungs were visually assessed and weighed for the calculation of mass coefficients.

Mice were examined for external pathological signs, condition of the chest and abdominal cavity, and internal organs. The internal organs (lungs and spleen) were removed and weighed (to calculate biometric indicators), and the material was taken for bacteriological examination, as well as macroscopic examination (with subsequent fixation in 10% neutral formalin solution).

The body weight of the model animals in both experimental series was monitored once a week. The organ weight coefficients for the lungs and spleen were calculated based on the ratio of the organ weight to the body weight of the animal. The index of

lung pathology was established based on the combined estimation of exudative and productive changes expressed in conventional units as previously described (Table S1) (20, 27).

The homogenized organs (0.1 g of lungs and whole spleen) were cultured on Loewenstein-Jensen medium (Becton Dickinson, USA) using the method of serial dilutions. The number of grown colonies of *M. tuberculosis* was counted after 4 weeks of incubation at +37°C. The growth of mycobacteria in cultures of lung homogenates was recalculated per organ's weight. Results were expressed as $\log_{10}$ CFU per organ.

All mice in this series that survived on day 196 p.i. were euthanized to assess the severity of the course of the tuberculosis process: lethality, lung weight coefficient, and lung pathology index. Since their increase correlates with the increasing severity of the lesion, these three values were summed up to obtain a cumulative average index (CAI) that served to compare the virulence of the studied strains.

Histological examination of the lungs was performed for animals of series 1 at the final stage of the experiment (112 days). After fixation, the lung pieces were placed in special plastic cassettes, labeled, and proceeded for histological preparation using an automatic processor Tissue-TekVip 5Jr (Sakura, Japan), embedding in paraffin (using Tissue-Tek AutoTEC a120 automated embedder), cutting (using "ThermoScientific" Microm HM43 microtome) followed by placement of sections on glass slides, deparaffinization, and staining of with hematoxylin and eosin. Microscopic examination was performed on a Nikon Ci-S microscope with video digital image processing for material archiving and further processing.

For statistical analysis, Microsoft Excel 2013 and Statistica 13.0 programs were used. The significance of differences was assessed by Student's *t*-test and was considered significant at $P < 0.05$. Lethality analysis of the model animals was performed using the Kaplan-Meier method. Pearson correlation coefficient (*r*) was calculated using https://www.statskingdom.com/correlation-calculator.html.

## RESULTS

### Genomic characterization of the studied strains

Twelve Russian LAM isolates of different spoligotypes were subjected to WGS and subsequent phylogenetic analysis (see Fig. 1 which also shows spoligotypes and resistance mutations). For the murine model study, we selected four LAM strains that were located in different parts of the tree. The isolates also differed in the drug resistance profile (see this and other information on strains in Table S2) that altogether correlated with drug resistance patterns commonly observed in strains of these spoligotypes (3–6, 11–14, 21). In particular, SIT264 is known to be mainly drug susceptible, SIT266 is XDR-associated, and SIT252 and SIT254 are MDR associated. Thus, these four LAM isolates represented the main trends in LAM population diversity in Russia: (i) sporadic susceptible strain (SIT264), (ii) potentially emerging MDR/pre-XDR strains (SIT252 and SIT266), and (iii) widespread drug-resistant strain (SIT254).

The susceptible reference strain H37Rv was also included in the murine study as a control. H37Rv is a laboratory virulent strain that was initially isolated from a TB patient in New York, USA in 1905 (28). Thus, all five strains studied in the murine model belong to different sublineages of Lineage 4 (Euro-American lineage): L4.3 (LAM) and L4.9 (H37Rv).

### Virulence study of C57BL/6 mice infected with reference and clinical isolates

The virulence of the strains was assessed by comparing the severity of the TB infection caused by different strains at days 7, 14, 21, 28, 56, and 112 p.i.

The changes in the lung and spleen weight coefficients reflect the development of specific inflammatory changes in these organs. Biometric examination of the lungs showed an overall increase in the lung weight coefficient in all groups of infected animals (Fig. 2A). On day 14 p.i., the lung weight coefficient increased significantly compared to day 7 p.i. ($P < 0.05$, $P < 0.002$) in four groups (Nos. 1, 2, 4, and 5). It was the

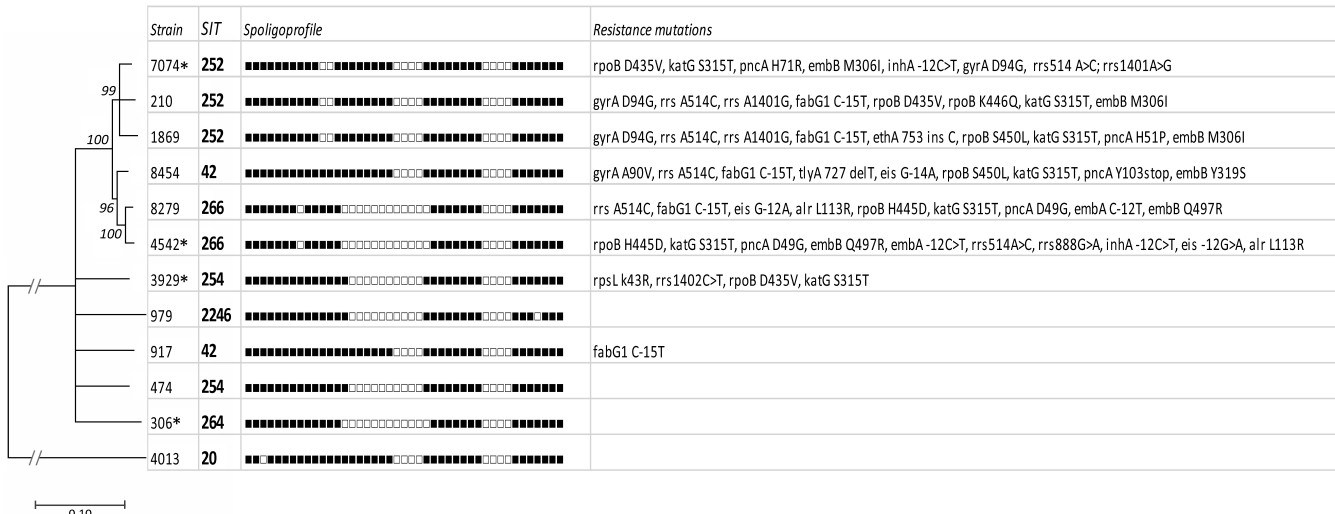

| Strain | SIT | Spoligoprofile | Resistance mutations |
|---|---|---|---|
| 7074* | 252 | | rpoB D435V, katG S315T, pncA H71R, embB M306I, inhA -12C>T, gyrA D94G, rrs514 A>C; rrs1401A>G |
| 210 | 252 | | gyrA D94G, rrs A514C, rrs A1401G, fabG1 C-15T, rpoB D435V, rpoB K446Q, katG S315T, embB M306I |
| 1869 | 252 | | gyrA D94G, rrs A514C, rrs A1401G, fabG1 C-15T, ethA 753 ins C, rpoB S450L, katG S315T, pncA H51P, embB M306I |
| 8454 | 42 | | gyrA A90V, rrs A514C, fabG1 C-15T, tlyA 727 delT, eis G-14A, rpoB S450L, katG S315T, pncA Y103stop, embB Y319S |
| 8279 | 266 | | rrs A514C, fabG1 C-15T, eis G-12A, alr L113R, rpoB H445D, katG S315T, pncA D49G, embA C-12T, embB Q497R |
| 4542* | 266 | | rpoB H445D, katG S315T, pncA D49G, embB Q497R, embA -12C>T, rrs514A>C, rrs888G>A, inhA -12C>T, eis -12G>A, alr L113R |
| 3929* | 254 | | rpsL k43R, rrs1402C>T, rpoB D435V, katG S315T |
| 979 | 2246 | | |
| 917 | 42 | | fabG1 C-15T |
| 474 | 254 | | |
| 306* | 264 | | |
| 4013 | 20 | | |

0.10

**FIG 1** WGS-based neighbor-joining tree of the Russian LAM isolates based on genome-wide SNPs (repeat regions excluded). LAM RD-Rio strain 4013 was used as an outgroup. All other isolates belong to the LAM-RUS branch. Isolates included in the murine model study are marked by an asterisk. The distance matrix for LAM-RUS strains is shown in Fig. S2.

highest at almost all periods of observation in mice infected with H37Rv and the lowest in mice infected with isolates 7074 and 3929. Detailed information on the significance of the pairwise comparisons between different strains with regard to different characteristics (lung and spleen weight coefficient, pathology index, and bacterial load) during the experiment is shown in Tables S3 through S12. The lung weight coefficient was significantly higher in group 5 (strain 4542, SIT266) compared to other clinical groups.

The pattern of changes in the spleen weight coefficient was similar in all groups and most pronounced at day 21 p.i. (Fig. 2B). Its significant increase in all groups was recorded on days 14 to 21 p.i. ($P < 0.05$, $P < 0.001$), followed by a steady decrease. The highest values, even slightly higher than H37Rv-group, were registered in group 5 (strain 4542, SIT266) at most of the time points.

The specific inflammation in the lungs assessed by the lung pathology index showed a similar increase in all groups of mice with higher scores in H37Rv and groups infected with strains 4542 (SIT266) and 306 (SIT264) (Fig. 3A) on days 14 and 21 p.i. and near the end of the experiment. Almost all animals from the experiment had single submiliary foci of productive tuberculous inflammation and small airless foci, pointing to the exudative component of the inflammatory reaction. The values of the pathology index were the highest in the H37Rv group, significantly exceeding those in other groups of mice at most of the time points ($P < 0.05$ to $P < 0.002$).

Multiple submiliary and single miliary productive foci were detected on day 14 p.i. in all experimental groups, leading to a significant increase in the pathology index ($P < 0.05$ to $P < 0.001$). Exudative manifestations of infection also increased, airless areas in mice of groups 1, 2, and 4 occupied half of the lungs, while single foci were in groups 3 and 5 and occupied half of the lungs only by day 21 p.i. The confluent nature of miliary productive foci was registered earlier (day 21 p.i.) in groups infected with strains H37Rv, 306, and 4542.

In all groups, a gradual increase in the lung pathology index was noted until day 28 p.i. when it reached almost the same level in all experimental animals, followed by a decrease (Fig. 3A). A closer look at the individual data revealed that this decrease was due to the reduced severity of the exudative component of the inflammatory reaction in clinical groups, manifested by decreased area of airless foci in the lungs. In the H37Rv group, two-thirds of the lungs were airless at days 56 and 112 p.i. Extensive airlessness of the lungs in group 1 (strain H37Rv) was observed in three mice on day 112 p.i. The highest values of the lung pathology index were recorded in mice infected with H37Rv,

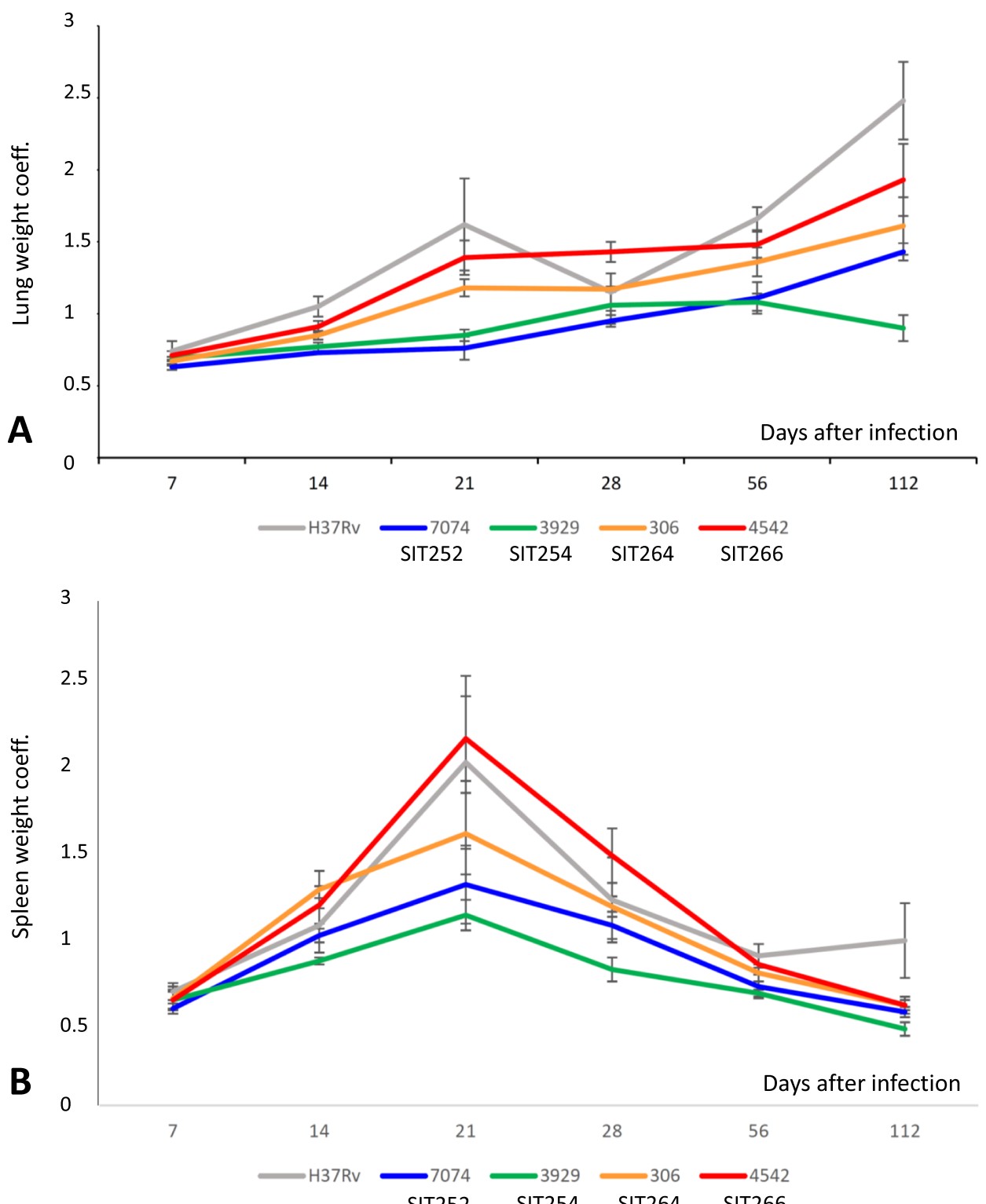

**FIG 2** Changes in the lung (A) and spleen (B) weight coefficients, in conventional units, of mice infected with *M. tuberculosis* strains determined at different time points. Here and in other figures, the same color is used to depict five studied strains. Data represent means plus standard deviations (SD) (error bars).

the lowest—in mice infected with strains 7074 (SIT252) and 3929 (SIT254) which showed similar changes. In turn, strains 306 (SIT 264) and 4542 (SIT266) showed intermediate and similar lung damage indices.

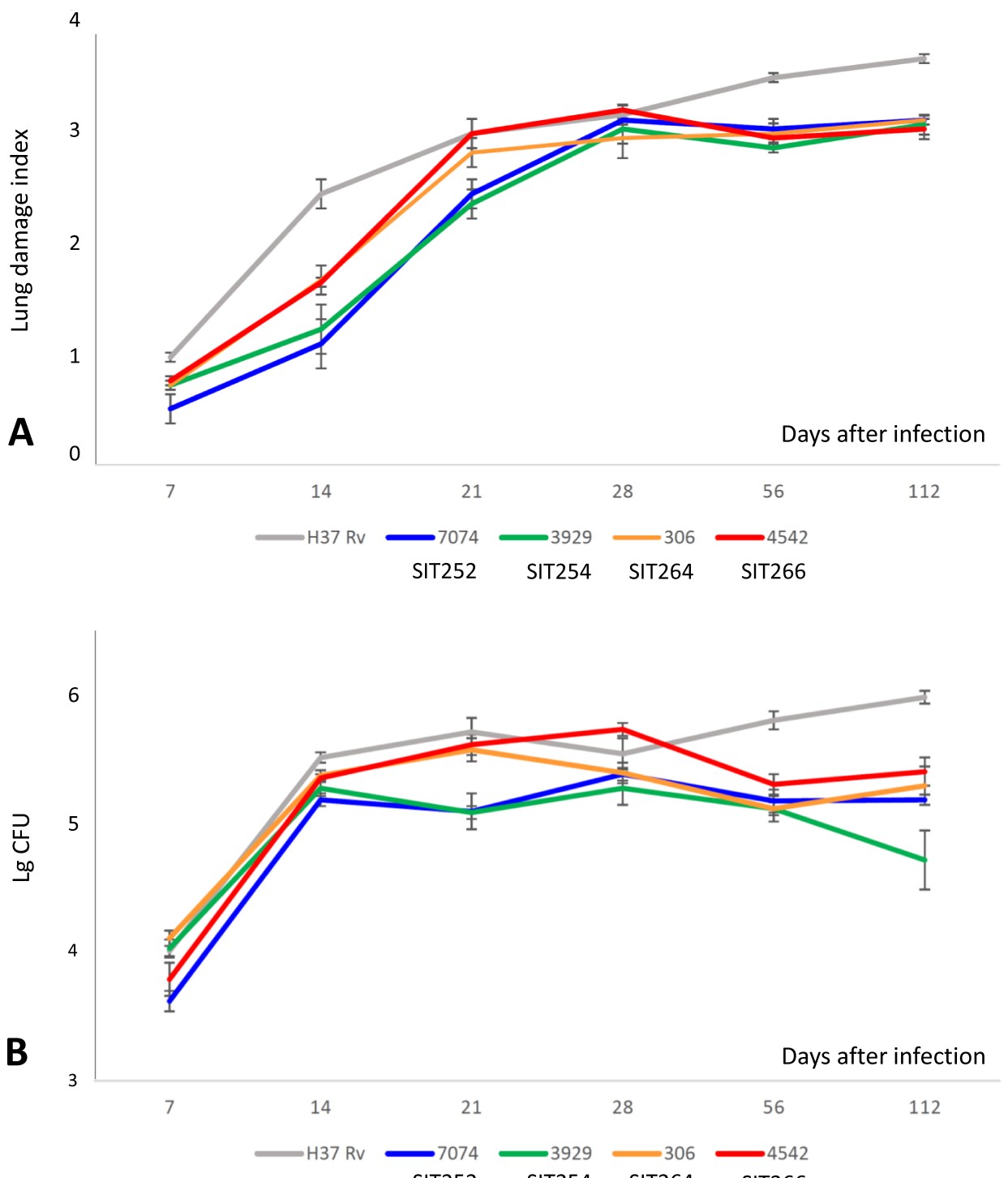

**FIG 3** Changes in lung pathology scores (A) and bacterial load in the lungs (B) of mice infected with *M. tuberculosis* strains determined at different time points.

The analysis of the bacterial load of the lungs confirmed the detected trends in the relative virulence of the studied strains (Fig. 3B). The lowest values were in group 2 (strain 7074, SIT252) and group 3 (strain 3929, SIT254). The highest bacterial load at most time points was noted in group 1 (H37Rv) and group 5 (strain 4542, SIT266). At day 21, the most contrasting levels of the bacterial loads were detected in mice infected with H37Rv, strains 306 and 4542 (similar and highest) vs mice infected with strains 7074 and 3929 (similar and lowest). At the end of the experiment (day 112 p.i.), the contrasting highest

and lowest bacterial loads were in the groups infected with H37Rv and 3929 (SIT254), respectively, while the other three strains were similar and intermediate.

An overall similar trend was observed for changes in the bacterial load of the spleen at day 21 p.i. (the higher in H37Rv, 306, 4252 vs the lower in 7974, 3929 infected groups of mice) (Fig. S3). Also, at day 112 p.i., the lowest bacterial load was in the 7074- and 3929-infected groups, the highest in the H37Rv-group, and intermediate in the groups infected with strains 306 and 4252. Interestingly, this indicator was similar in all five groups at day 14 p.i. followed by its overall and clear decrease. This differed from much less pronounced changes in the bacterial load of the lungs (Fig. 3B).

The correlation between bacterial load and lung pathology index was estimated by Pearson correlation coefficient ($r$) (Table S13). In most cases, it was non-significant: very small negative for 7074 (SIT252) and 3929 (SIT254), and large positive for 306 (SIT264) and H37Rv. Only for one strain 4542 (SIT266), a significant large positive relationship was demonstrated.

## Histological evaluation of lung sections

Histological examination demonstrated that all infected mice developed extensive lung damage with strain-specific features. Two types of lung lesions were observed: more widespread lung damage caused by strains H37Rv and 306, and less widespread lung damage caused by strains 7074, 3929, and 4542. We tentatively termed these states as Type I and Type II lesions, respectively. These types also differed in the nature of lesions. Predominantly, alterative-exudative changes were observed for Type I while mainly productive changes were characteristic for Type II as detailed below (Fig. 4; Fig. S10).

Type I changes in the lungs included large foci of infiltration without clear boundaries, merging into conglomerates (Fig. 4A). At the same time, the airiness of the lung tissue (i.e., the ratio of the aired areas to the total area of the section) decreased by >30% in all mice infected with H37Rv, and in four out of six mice infected with strain 306 (Table S14). The interalveolar septa in the infiltration zones were noticeably thickened due to the accumulation of neutrophilic granulocytes, lymphocytes, and macrophages (Fig. S4). These cells were also located in the lumen of the alveoli, where their microscopic features were most clearly revealed. First of all, these were foamy macrophages and epithelioid cells (Fig. S5). In some alveoli, the infiltrate consisted only of these cells, closely adjacent to each other, while in others, the cells were located loosely and with serous exudate between them. There were also alveoli completely filled with serous exudate without

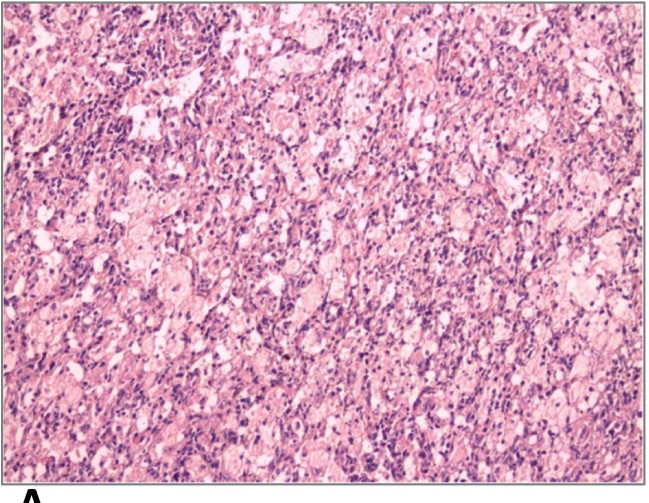
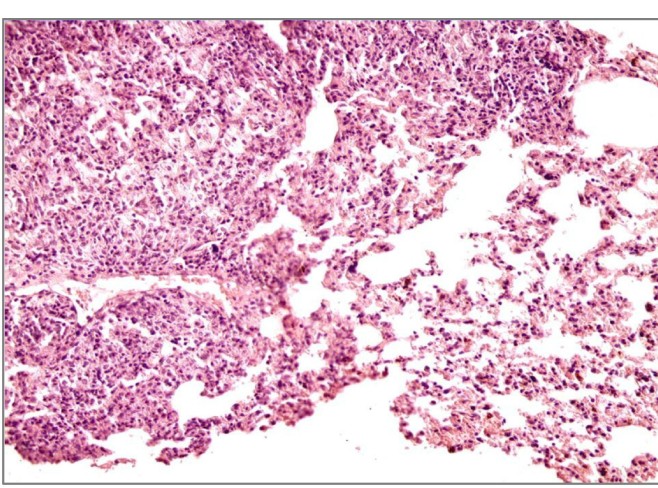

**A.**
**B.**

**FIG 4** Microscopic images at day 112 p.i. of the lung of mice infected with (A) *M. tuberculosis* H37Rv; note the confluent foci of specific infiltration without clear contours. (B) Strain 4542; note the focus of specific infiltration surrounded by air tissue. Stained with hematoxylin and eosin ×300.

admixture of cells, as well as alveoli containing neutrophilic granulocytes. In 4 out of 12 mice, mild perivascular and peribronchial lymphohistiocytic infiltration was determined in the lungs.

Type II changes in the lungs were characterized by clearly defined small airless areas surrounded by air tissue (Fig. 4B). A decrease in airiness by >30% was observed only in one mouse infected with strain 3929 (Fig. S6), and in two mice infected with strain 4542. In general, these changes were three times less abundant compared to the Type I changes (Table S14).

Type I and II changes also differed in the cell content of infiltrates. Thus, type II changes were predominantly productive and consisted mainly of lymphocytes, foamy macrophages, and epithelioid cells (Fig. S7). A small number of alveoli were filled with serous exudate, while neutrophilic granulocytes were found in lymphohistiocytic clusters only in one mouse infected with strain 3929 and in two mice infected with strain 4542 (Fig. S8). Finally, peribronchial and perivascular lymphohistiocytic aggregations observed in 18 mice were another striking histological characteristic of the productive response in type II changes (Fig. S9 and S10).

## Survival study of C57BL/6 mice infected with reference and clinical isolates

Survival rates differed significantly between the studied groups of mice infected with different strains (Fig. 5). Mice infected with strains 7074, 3929, and 4542 had the lowest lethality that did not exceed 5% and was recorded only after day 150 p.i. By contrast, the death of mice in the groups infected with H37Rv and 306 was recorded on day 21 p.i. Survival curves in these groups followed a similar course up to day 116 p.i. with ~20% of died animals but started to change in days 117–174 p.i. with 100% lethality in H37Rv group and 50% lethality in the 306 group. Thus, in terms of survival, the studied strains can be divided into three types: high 100% lethality (H37Rv), moderate 50% lethality (strain 306, SIT264), and low 5% lethality (strains 7074, 3929, and 4542).

An analysis of the body weight of mice showed that groups infected with 7074, 3929, and 4542 had the highest average weight values, indicating the most favorable course

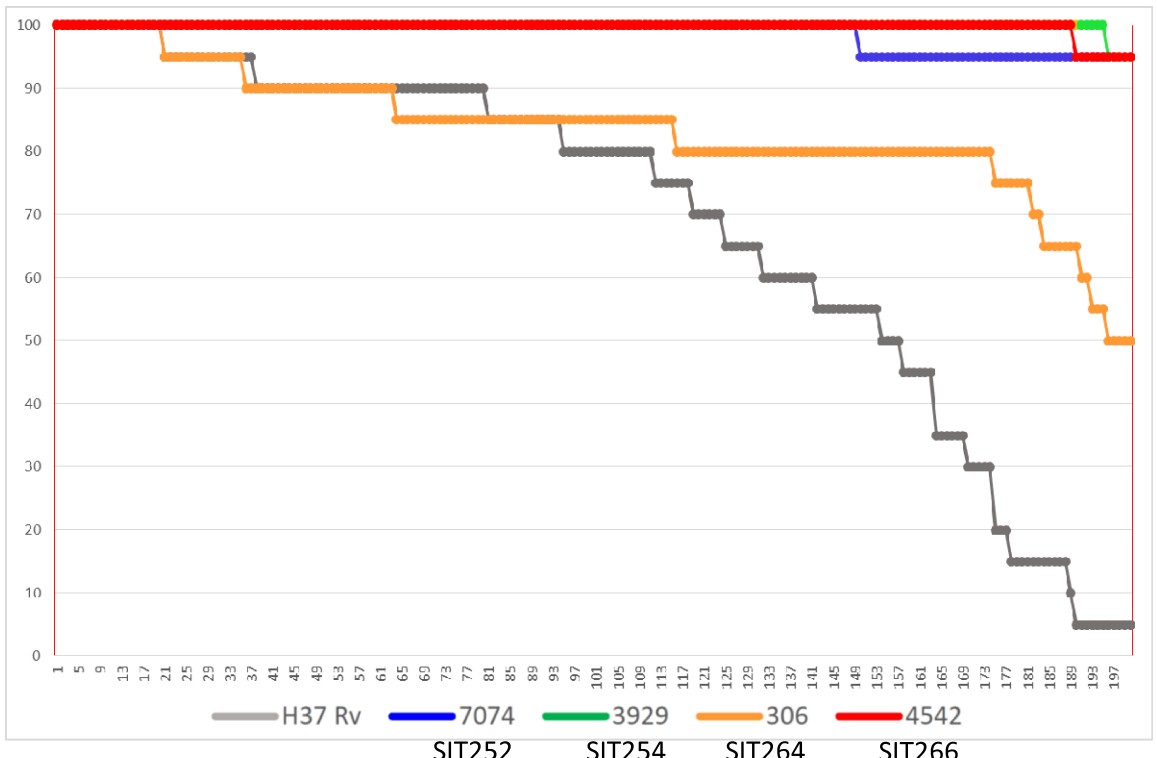

**FIG 5** Survival of mice after infection with *M. tuberculosis* strains.

of infection. The lowest body weight was in mice infected with strains H37Rv and 306 (Fig. S11). A certain decrease in the body weight in mice infected with strain 306 was noted after the 14th week of the experiment, which reflects the pronounced intoxication syndrome.

A CAI was used to compare the virulence of the studied strains (Table S15). The highest value of this index (19.06) was in mice infected with strain 306 (SIT264) due to the high lung weight coefficient and lung damage. The other three clinical strains showed similar and much lower values (3.3–3.6). The group infected with strain 7074 (SIT252) had the lowest cumulative index due to concomitantly lowest lethality, lung weight coefficient, and lung damage.

## Biologically meaningful mutations in strains 7074 vs 4542

This virulence study focused on four LAM isolates that represent phylogeographically and genomically distinct medically relevant spoligotypes. An in-depth genomic study of the SIT-specific genetic variation requires analysis of the enlarged collection of isolates (that is ongoing). Preliminarily, we assessed the available genomic data of the genetically closely related MDR isolates with contrasting virulence characteristics: low virulent strain 7074 (SIT252) and highly virulent strain 4542 (SIT266). In total, 13 strain-specific biologically meaningful mutations were found (Table S16).

## DISCUSSION

Current views on the interplay between virulence, fitness, resistance, and transmissibility of *M. tuberculosis* strains suggest that (i) multiple resistance mutations reduce fitness and transmission capacity which explains the selection of the compensatory mutations in some successful strains; (ii) highly virulent strains are also highly transmissible; and (iii) the most successful strains harbor both low-cost resistance and/or compensatory mutations that restore their fitness (29–32). The findings of this study are partly in line with these observations. The studied isolates belong to different and epidemiologically significant subtypes of the LAM-RUS branch of *M. tuberculosis*. These are (i) the evolutionarily more recent and possibly emerging MDR SIT252 and SIT266, (ii) geographically diverse and drug-susceptible SIT264, and (iii) SIT254 that is the commonest LAM spoligotype in Russia and neighboring countries. Spoligotypes SIT264 and SIT266 are very similar and differ in one signal #8 that is invisible in SIT266 due to asymmetrical IS*6110* insertion (12). Noteworthy, WGS analysis showed that these SIT are not directly related and differ in 92 SNPs (Fig. 1).

To the best of our knowledge, only a few articles on the LAM virulence in the murine model were published, all were in the Russian language and were published more than 10 years ago (15, 16). On the other hand, previous virulence studies of the Euro-American lineage enrolled other than LAM genotypes. For example, a study of clade-specific virulence patterns in human primary macrophages and mice infected by the aerosol route demonstrated high uptake, high cytokine induction, and high growth rates for Haarlem strains (33). It is unknown whether LAM strains would exhibit similar or partly similar properties.

## Different patterns of the key virulence and lethality features

The main results on characteristic virulence features of different strains are summarized in Fig. 6 where the studied strains are listed in the descending order of their relative virulence and lethality. It appears that there is a non-straightforward relationship between different characteristics and different types of correlation may be seen. For example, pre-XDR strain SIT252 appears the least virulent while pansusceptible strain SIT264 is the most virulent, lethal, and leading to the most extensive histological changes in the lungs. The higher number of resistance mutations is in negative correlation with virulence in SIT252 but the situation was more complex with strain SIT266 which is both the most virulent and bears 10 drug resistance mutations. Finally, the highest virulence

| Strain, spoligotype, resistance | Lung weight coefficient | Spleen weight coefficient | Lung pathology index | Bacterial load of the lungs | Bacterial load of the spleen | Histological examination of the lung sections | Lethality | Cumulative average index (based on Survival series, at day 196 p.i.) |
|---|---|---|---|---|---|---|---|---|
| H37Rv, susceptible | High and highest in most points | High (highest at day 112) | High. Airless areas - 50% at day 14 | High, days 21, 112 | Highest | Most widespread, mainly alterative-exudative changes | High | Not assessed |
| 306, SIT264, susceptible | intermediate | intermediate | Intermediate. Airless areas - 50% at day 14 | High, day 21; Intermediate, day 112 | Intermediate | More widespread, mainly alterative-exudative changes | Intermediate | Highest (due to the high lung weight coefficient and lung damage) |
| 4542, SIT266, MDR, 10 resistance mutations | Higher than other isolates | Highest | Intermediate. Airless areas - 50% | High, day 21; Intermediate, day 112 | Intermediate | Less widespread, mainly productive changes | Low | Low |
| 7074, SIT252, Pre-XDR, 9 resistance mutations | Lower | Low | Low Single foci day 14; Airless areas - 50% at day 21 | Low, day 21; Intermediate, day 112 | Low | Less widespread, mainly productive changes | Low | Lowest (due to the lowest lethality, lung weight coefficient and lung damage) |
| 3929, SIT254, MDR, 4 resistance mutations | Lower, Lowest | Low, lowest | Low. Single foci at day 14; Airless areas - 50% at day 21 | Low, day 21; Lowest, day 112 | Low, lowest | Least widespread, mainly productive changes | Low | Low |

**FIG 6** Summary of the main virulence and lethality characteristics of the studied strains. Observations are mostly for the period between days 14 and 56 p.i. The cumulative average index was based on lethality, lung weight coefficient, and lung damage of survived mice at 196 dpi within the survival series. In addition, the high, intermediate, and low values are marked by color (red, orange, and blue, respectively).

and lethality of all studied strains were observed for strain H37Rv, a control virulent strain.

The histological analysis corroborated biometric results (bacterial load, pathology index). It showed that characteristic inflammation developed in the lungs of mice by the day 112 p.i. was different for different strains in terms of abundance and severity (Fig. 4; Fig. S9), thus correlating with different virulence of the strains. The most virulent strains were H37Rv and susceptible 306 (SIT264), which caused widespread pneumonic confluent changes in the lungs (that we defined as Type I changes). At the same time, strains 7074, 3929, and 4542 caused discrete small-focal lung lesions (Type I changes) and can be considered relatively less virulent. This conclusion is also supported by the microscopic picture, with a clear predominance of an alterative-exudative reaction for strains H37Rv and 306, compared to the clearly proliferative manifestation for three other strains. It may be that more virulent strains H37Rv and 306 multiply at a higher rate and accumulate more abundantly in the lungs, as evidenced by the abundance of foamy macrophages containing mycobacterial antigens.

The less virulent strains 7074, 3929, and 4542 are interesting in terms of their immunogenic properties. Indeed, in the lungs of mice infected with these strains, characteristic perivascular and peribronchial lymphohistiocytic infiltrates play a major role in the immune response. Similar reactive changes developed in the lungs during experimental infection with the BCG vaccine strain, ensuring its survival and limited reproduction in the animal body with minimal damaging effects on tissues (34).

Interestingly, in line with our findings of the striking pathogenetic differences between the closely related strains, the same was shown for LAM RD-Rio isolates in Colombia that were compared in their transcriptional response under two axenic media conditions (35). Those clinical isolates were found phenotypically different at the level of cell death, cytokine production, growth kinetics upon *in vitro* infection of human tissue macrophages, and membrane vesicle secretion upon culture in synthetic medium. Furthermore, RNA-seq analysis has identified different strategies to counteract the adverse condition of a carbon-poor media: (i) activation of virulence systems such as the ESX-1, synthesis of diacyl-trehalose, polyacyl-trehalose, and sulfolipids and (ii) activation of the DNA replication, cell division, and lipid biosynthesis (35).

## Pathobiologically meaningful mutations suggested by *in silico* analysis

A comparison of WGS data for low-virulent strain 7074 (SIT252) and highly virulent strain 4542 (SIT266) revealed eight potentially biologically meaningful mutations for strain 7074 and five such mutations for strain 4542 (Table S16). Hypothetically, the former could be associated with reduced virulence while the latter—with increased virulence, respectively. Indeed, previous studies suggested the role of some of these genes/proteins in the virulence of *M. tuberculosis* or other pathogens (36–50) (Table S16). The following mutated proteins are of particular interest: (i) TatD DNase potential virulence factor in *Plasmodium falciparum* and *Streptococcus pneumoniae* involved in biofilm formation and required for virulence of *Trueperella pyogenes* (48), (ii) glycogen phosphorylase GlgP abundant after phagocytosis of K-strain dominant in Korea (44), and (iii) carboxylesterase A CaeA (Rv2224c) modulates innate immune response and is required for full virulence (40). In *T. pyogenes,* biofilms formed by TatD mutants produced a lower bacterial load in the spleen of mice and compromised virulence (48). In our study, a mutation TatD C11R in the SIT252 strain could also contribute to its reduced virulence manifested by low values of all assessed characteristics (Fig. 6 ). Similarly, *Rv2224c* (*CaeA*), a cell envelope-associated predicted protease, is critical for *M. tuberculosis* virulence. Disruption of *Rv2224c* led to prolonged survival of infected mice and highly reduced lung pathology (43). The absence of *Rv2224c* enhanced host innate immune responses and compromised the intracellular survival of *M. tuberculosis* in macrophages. While no *pks8* studies were published on *M. tuberculosis*, the polyketide synthases are known as virulence factors and Pks8 was shown to be required for the pathogenicity of plant pathogen *Pseudocercospora fijiensis* (46). The true role of these mutations should be further determined in the experimental model, for example, using allele replacement experiments or knockout mutants.

## Dynamic changes in the phylogeographic landscape

We further tried to assess the dynamic changes in the phylogeography of these four spoligotypes based on the studies published in the last 20 years from different Russian regions, Belarus and Ukraine (5, 21, 51–60) (Fig. 7). SIT252 was previously termed as emerging since it included MDR-associated, genetically closely related and geographically delimited isolates from European Russia (14); in our study, it was represented MDR and low-virulent strain 7074. Surprisingly, SIT252 was not described in the studies published after 2014. In contrast, rare and drug-susceptible SIT264 (virulent and moderately lethal strain 306 in the present study) slightly increased its circulation and geography. Moderately drug-resistant SIT254 (the least-virulent strain 3929 in this study) remained the most widespread LAM spoligotype. A certain caution in the interpretation of the subtype prevalence is however required. While most of the studies (all in Fig. 7B and most in Fig. 7A) were population based, some earlier studies presented in Fig. 7A were based on convenience or small samples (St. Petersburg, Yakutia, Kazakhstan) or included only drug-resistant strains (Belarus), and thus could be biased.

The situation with susceptible and virulent SIT264 that is on the increase in the European part of Russia is similar to the increase in susceptible Beijing strain in Cape Town, South Africa which became visible under a longitudinal study over 12 years. The incidence of the Haarlem, LAM, Quebec, and the Low-Copy Clades remained relatively stable, in contrast to the exponentially increased incidence of the Beijing strains. This growth was exclusively attributable to drug-susceptible Beijing strains that had a greater proportion of smear-positive sputa than their non-Beijing counterparts and were less likely to be successfully treated (61). These differences likely reflected enhanced pathogenicity rather than transmissibility of the Beijing genotype in Cape Town.

## Limitations

The limitation of the mice experiments comparing different genotypes of *M. tuberculosis* is the relatively small number of strains that can be investigated in each experiment. Only

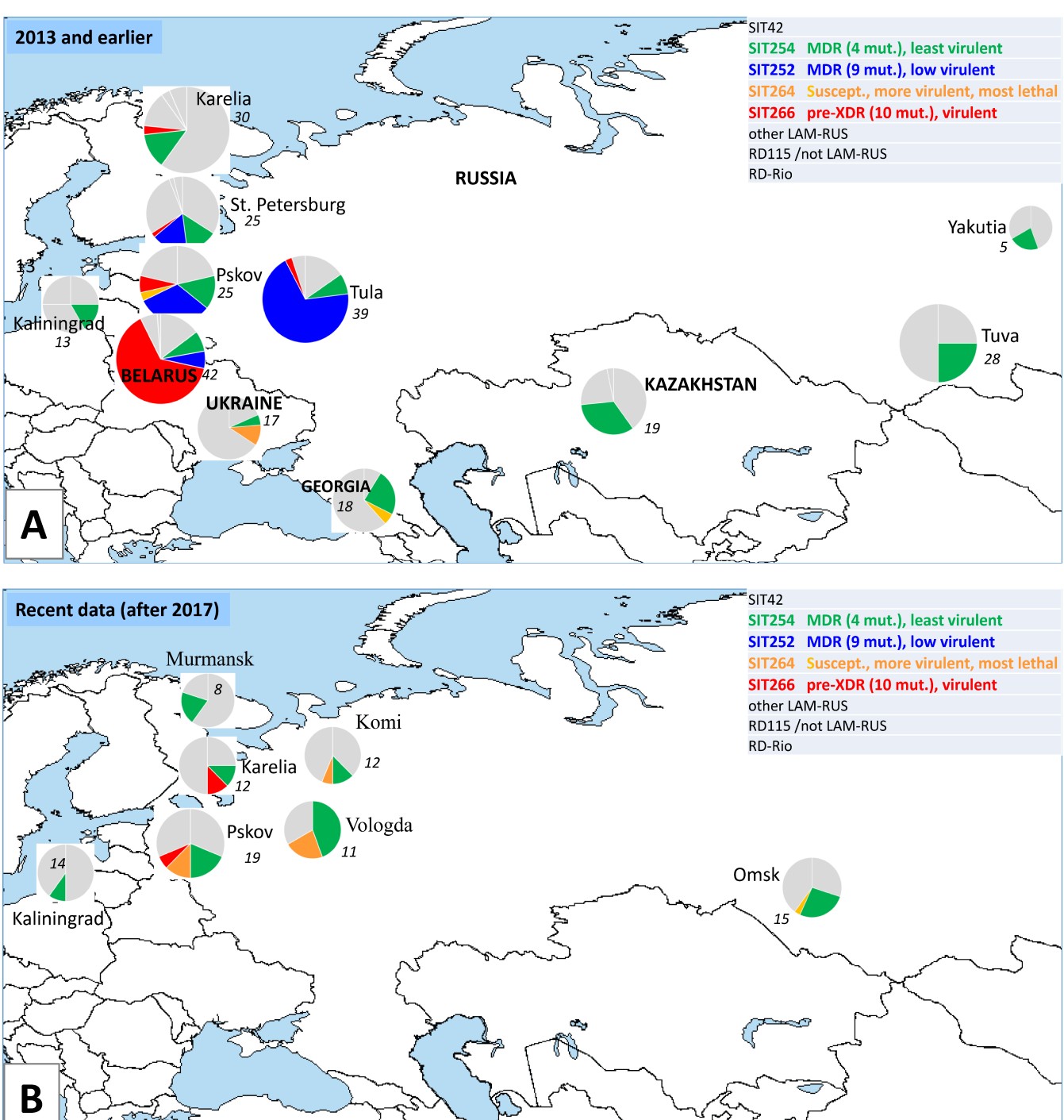

**FIG 7** LAM subtypes: geography, resistance, and relative virulence in earlier (A) and recent studies (B). Circle size is roughly proportional to the percentage of LAM in a local sample (also shown in italics). The original map, freely available at https://de.wikipedia.org/wiki/Datei:A_large_blank_world_map_with_oceans_marked_in_blue.PNG, is licensed under the Creative Commons Attribution-Share Alike 3.0 Unported License. Pie charts, data labels, and legends have been overlaid by the authors.

one isolate per spoligotype was included in this study. However, this limitation is almost inevitable for this kind of *in vivo* studies that are difficult to include multiple isolates given the already large sample size of the studied animals. Nonetheless, we believe that the enrolled clinical isolates correctly represent their spoligotypes in terms of phenotypic resistance features. The used model of intravenous tail injection was previously shown by

us and others as a reliable method to study the virulence and lethality of the *M. tuberculosis* isolates (20, 23–25).

## Conclusion

Putting together genomic, phenotypic, and virulence features of the LAM strains and more generally, dynamic phylogeography of their respective spoligotypes over 20 years, revealed a peculiar and partly inverse correlation and three types of manifestation. First, the most virulent, more lethal but drug-susceptible strain SIT264 increased its circulation in parts of Russia. Second, virulent and pre-XDR strain SIT266 was prevalent in Belarus and continues to be visible in North-West Russia. Third, low-virulent and MDR strain SIT252 with multiple resistance mutations was widespread in European Russia 10 years ago but drastically reduced its current circulation. The former two cases appear to be the most hazardous and require particular attention.

These observations emphasize strain virulence as an important feature associated with transmission, irrespective of drug resistance properties. Our findings suggest that increased virulence of some of the clinical isolates is associated with a fundamentally different systemic immune response, which can be detected early during infection. Virulence may correlate with the rate of reproduction of microbes in a given system (humans, mice), and if the notorious MDR strains multiply slowly, the immune system would have enough time to counteract them.

The studied strains while heterogeneous in their pathobiological features, belong to the same LAM-RUS branch, a part of the LAM family, itself a member of Lineage 4. However, Lineage 4 is heterogeneous and its other genotypes (Haarlem, Ural, S, etc.) have a different interplay of bacterial virulence and resistance, which was also modulated by coevolution with local human populations. Different lineages and sublineages, and smaller more homogeneous clusters of human *M. tuberculosis* isolates have developed their way of adaptation and dissemination.

Medically relevant research should focus on genetically compact epidemic clusters. Increasing circulation of the susceptible but virulent and lethal strains implies (i) the importance of their close epidemiological monitoring and (ii) that personalized TB treatment should consider not only resistance but also the virulence of the infecting strains. This highlights importance of development of the so called anti-virulence drugs.

## ACKNOWLEDGMENTS

This study was supported by Russian Science Foundation (grant 19-14-00013).

Conceptualization: I.M., T.V. Investigation: M.D., N.Z., A.V., M.V., N.S. Formal analysis: I.M., T.V., B.A. Writing, original draft: I.M. Writing - Review & Editing: I.M., T.V., B.A.

Authors declare that no conflict of interest exists.

## AUTHOR AFFILIATIONS

[1]St. Petersburg Pasteur Institute, St. Petersburg, Russia
[2]St. Petersburg Research Institute of Phthisiopulmonology, St. Petersburg, Russia

## AUTHOR ORCIDs

Igor Mokrousov http://orcid.org/0000-0001-5924-0576

## FUNDING

| Funder | Grant(s) | Author(s) |
| --- | --- | --- |
| Russian Science Foundation (RSF) | 19-14-00013 | Igor Mokrousov |
| | | Tatiana Vinogradova |
| | | Anna Vyazovaya |

## AUTHOR CONTRIBUTIONS

Igor Mokrousov, Conceptualization, Formal analysis, Writing – original draft, Writing – review and editing | Tatiana Vinogradova, Conceptualization, Formal analysis, Writing – review and editing | Marine Dogonadze, Investigation | Natalia Zabolotnykh, Investigation | Anna Vyazovaya, Investigation | Maria Vitovskaya, Investigation | Natalia Solovieva, Investigation | Boris Ariel, Formal analysis, Writing – review and editing

## DATA AVAILABILITY

All data of this study are presented in the article and supplementary material. Data for the *M. tuberculosis* genomes were deposited in the NCBI Sequence Read Archive (project number PRJNA886055).

## ADDITIONAL FILES

The following material is available online.

### Supplemental Material

**Supplemental figures (Spectrum01392-23-s0001.pdf).** Fig. S1 to S11.
**Supplemental tables (Spectrum01392-23-s0002.pdf).** Tables S1 to S16.

### Open Peer Review

**PEER REVIEW HISTORY (review-history.pdf).** An accounting of the reviewer comments and feedback.

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
