## [Reviewer comments · Microbiology Spectrum]

Microbiology Spectrum

A multifaceted interplay between virulence, drug resistance and phylogeographic landscape of *Mycobacterium tuberculosis*

Igor Mokrousov, Tatiana Vinogradova, Marine Dogonadze, Natalia Zabolotnykh, Anna Vyazovaya, Maria Vitovskaya, Natalia Solovieva, and Boris Ariel

Corresponding Author(s): Igor Mokrousov, St. Petersburg Pasteur Institute

Review Timeline:

Submission Date:	March 31, 2023
Editorial Decision:	June 7, 2023
Revision Received:	July 6, 2023
Accepted:	August 6, 2023

Editor: Alexandra Aubry

Reviewer(s): The reviewers have opted to remain anonymous.

Transaction Report:

DOI: <https://doi.org/10.1128/spectrum.01392-23>

June 7, 2023

Dr. Igor Mokrousov
St. Petersburg Pasteur Institute
Laboratory of Molecular Epidemiology and Evolutionary Genetics
14 Mira street
St. Petersburg 197101
Russia

Re: Spectrum01392-23 (A multifaceted interplay between virulence, drug resistance and phylogeographic landscape of Mycobacterium tuberculosis)

Dear Dr. Igor Mokrousov:

Link Not Available

Sincerely,

Alexandra Aubry

Journals Department
Reviewer comments:

Reviewer #1 (Comments for the Author):

The manuscript by Mokrousov et al deals with the assessment of virulence of Mycobacterium tuberculosis strains. It combines the use of the animal model of C57BL/6 mice to the exploration of disease dynamics in humans. The choice of strains to be studied further was done according to a phylogenetic study using WGS data. Distant strains were chosen to maximize the chance for showing different phenotypes by selecting multidrug resistant isolates and one susceptible. The study deals with a very low number of strains from a specific lineage of M. tuberculosis strains impeding large conclusions on whether the animal model can reliably help characterize the virulence of pathogenic strains, but it interestingly draws back attention to the necessity

of focusing on virulence and not solely on resistance. In addition, focusing on a relatively small relatively undersequenced sublineage, it allows to formulate new hypotheses regarding the evolution of this sublineage (LAM-RUS). The methods are globally appropriate and the gathered data are clearly presented. Despite the global relevance of the study, some points need to be improved before publication:

Major points :

-distribution between results and discussion is sometimes irrelevant:

*Results related to SNPs specific of different strains (table S15) should belong to results section, even if the significance of these results may be low. Accordingly, corresponding methods should be added in the methods section.

*The same is true for results related to Fig. 6. Regarding these data, it is important to specify on what collections they rely and whether these collection could be biased.

*In contrast to these two subpoints part of which should move from discussion to results section, limitations paragraph (currently in results section) belongs to discussion.

-the phylogenetic analysis is not performed in a way that allows to assess correctly the relative distance between strains: the tree of Fig. 1 lacks an outgroup to ensure proper rooting. Please rebuild the tree including an outgroup and perform bootstrap analyses. Also, basic information should be added in the legend: number of SNPs in the matrix used to build the tree, tree scale, bootstrap support of the main branches at least. It is important to mention whether the tree is derived from SAM-TB or Phylresne SNPs inference. Also, please specify whether resistance loci were included or not in the building of the tree.

Minor points:

-please provide deposit SRA onto a public platform and present accession number

-I.137 : please specify the time and place of sampling

-I. 206: specify what was taken as a reference for weight coefficient and mention the range of weights for the reference point.

-I.251: rephrase "drug resistance properties of their spoligotypes": a spoligotype has no resistance properties per se (but you may mention correlations between these features as specified shortly after)

-I. 267: specify what difference was found significant: day14 as compared to day7? And refer to strains not to group numbers or use numbers in association to infecting strains in the figure and in the first mention in the text.

-I.372 and following: some info on the severity index used could be transferred to methods and values should be shown in a supplemental table or all provided in the text.

-I. 403 to 409, discussion on evolution pathways of SITs is of little interest to the study and could be reduced

-I. 425 More number of resistance mutations -> change for more resistance mutations

-Fig. 2A, correct colour code (red line in the legend for 4542 strain)

-p.i. upon first use, specify abbreviation

Staff Comments:

Preparing Revision Guidelines

Please return the manuscript within 60 days; if you cannot complete the modification within this time period, please contact me. If you do not wish to modify the manuscript and prefer to submit it to another journal, please notify me of your decision immediately so that the manuscript may be formally withdrawn from consideration by Microbiology Spectrum.

Spectrum01392-23

A multifaceted interplay between virulence, drug resistance and phylogeographic landscape of *Mycobacterium tuberculosis*

Igor Mokrousov et al.

ANSWERS TO REVIEWER

Dear Reviewer, thank you very much for comments and critiques. We addressed all your comments in the revised version of our manuscript. The detailed answers are below. We also provide a manuscript with marked-up changes as a supporting file (for review only).

Reviewer #1 (Comments for the Author):

The manuscript by Mokrousov et al deals with the assessment of virulence of *Mycobacterium tuberculosis* strains. It combines the use of the animal model of C57BL/6 mice to the exploration of disease dynamics in humans. The choice of strains to be studied further was done according to a phylogenetic study using WGS data. Distant strains were chosen to maximize the chance for showing different phenotypes by selecting multidrug resistant isolates and one susceptible. The study deals with a very low number of strains from a specific lineage of *M. tuberculosis* strains impeding large conclusions on whether the animal model can reliably help characterize the virulence of pathogenic strains, but it interestingly draws back attention to the necessity of focusing on virulence and not solely on resistance. In addition, focusing on a relatively small relatively undersequenced sublineage, it allows to formulate new hypotheses regarding the evolution of this sublineage (LAM-RUS). The methods are globally appropriate and the gathered data are clearly presented. Despite the global relevance of the study, some points needs to be improved before publication:

Major points:

-distribution between results and discussion is sometimes irrelevant:

*Results related to SNPs specific of different strains (table S15) should belong to results section, even if the significance of these results may be low. Accordingly, corresponding methods should be added in the methods section.

ANSWER: Only first part of this paragraph is suitable for Results. Otherwise, Table S15 and further text present a discussion based on the published studies (Page 19).

Accordingly, we moved part of this section to Results (Page 16, last para), and also added briefly to Methods that “As potentially biologically meaningful mutations, we defined those with significant SIFT *P* value or short indels. We also considered mutations in the putative promoter regions.” (Page 7, lines 168-170). Otherwise, SNPs were detected using SAM-TB pipeline that we cited.

*The same is true for results related to Fig. 6. Regarding these data, it is important to specify on what collections they rely and whether these collection could be biased.

ANSWER: The used studies (that we cited: [5,21,51-60]) were population-based or convenience samples. This is added in revision: “A certain caution in interpretation of the subtype prevalence is however required. While most of the studies (all in Fig. 6B and most in Fig. 6A) were population-based, some earlier studies presented in Fig. 6A were based on convenience or small samples (St. Petersburg, Yakutia, Kazakhstan) or included only drug resistant strains (Belarus), and thus could be biased.” (Page 20, lines 488-492).

*In contrast to these two subpoints part of which should move from discussion to results section, limitations paragraph (currently in results section) belongs to discussion.

ANSWER: done as suggested (Page 21).

-the phylogenetic analysis is not performed in a way that allows to assess correctly the relative distance between strains: the tree of Fig. 1 lacks an outgroup to ensure proper rooting. Please rebuild the tree including an outgroup and perform bootstrap analyses. Also, basic information should be added in the legend: number of SNPs in the matrix used to build the tree, tree scale, bootstrap support of the main branches at least. It is important to mention whether the tree is derived from SAM-TB or Phyresse SNPs inference. Also, please specify whether resistance loci were included or not in the building of the tree.

ANSWER: we added an outgroup strain of the other and distant LAM sublineage (RD-Rio) and rebuilt the ML tree (see revised Figure 1, with scale and bootstrap values). Yes, we used only SAM-TB online tool, and under its pipeline, PE/PPE and resistance genes are excluded from phylogenetic analysis (this is added to revised Methods section) (Page 6, lines 150-151). We also provide a distance matrix for LAM-RUS strains as new supplementary Figure S2.

Minor points:

-please provide deposit SRA onto a public platform and present accession number

ANSWER: PRJNA was provided in M&M (lines 152-153 in the original ms): “Data for the *M. tuberculosis* sequenced genomes were deposited in the NCBI Sequence Read Archive (project number PRJNA886055).” (Page 7, lines 150-151).

-l.137 : please specify the time and place of sampling

ANSWER: Information on strain, origin or source was shown in Table S2 and is available under SRA deposited data for all strains. In revised paper, we added year of isolation in Table S2.

-l. 206: specify what was taken as a reference for weight coefficient and mention the range of weights for the reference point.

ANSWER: In M&M, we provided a detailed description on this calculation (Page 9, lines 210-212). In Results, Figure 2, we show these values for lung and spleen weight coefficients as means with SD (error bars).

-l.251: rephrase "drug resistance properties of their spoligotypes": a spoligotype has no resistance properties per se (but you may mention correlations between these features as specified shortly after)

ANSWER: this sentence was rephrased for clarity and references were added:

“The isolates also differed in the drug resistance profile (see this and other information on strains in Table S2) that altogether correlated with drug resistance patterns commonly observed in strains of these spoligotypes [3-6,11-14,21]. In particular, SIT264 is known to be mainly drug-susceptible, SIT266 is XDR-associated, SIT252 and SIT254 are MDR-associated.” (Page 11, lines 259-260).

-l. 267: specify what difference was found significant: day14 as compared to day7? And refer to strains not to group numbers or use numbers in association to infecting strains in the figure and in the first mention in the text.

ANSWER: yes compared to day 7. The sentence is revised (Page 12, lines 275-276). Regarding providing strain numbers along with SIT, they are shown in all figures except for this technical omission in Fig. 2A. We corrected.

-l. 372 and following: some info on the severity index used could be transferred to methods and values should be shown in a supplemental table or all provided in the text.

ANSWER: this paragraph was revised as you suggested (Page 16, lines 378-382). Its first sentences were moved to Methods (Page 10, lines 233-237). Table 1 was moved to Supplement, as new Table S15.

-l. 403 to 409, discussion on evolution pathways of SITs is of little interest to the study and could be reduced

ANSWER: We shortened this part from 6 to 3 lines (page 17, lines 405-407).

-l. 425 More number of resistance mutations -> change for more resistance mutations

ANSWER: I am sorry but not sure I understood correctly. An assumption suggested long ago, is that more changes in drug resistance genes (some encode for the key proteins, e.g rpoB – for RNA polymerase beta-subunit) would counteract drug action but would reduce strain fitness hence transmission. Multiple such mutations will theoretically reduce fitness (and virulence) – as we speculatively show for strain 7074 / SIT252. However, some successful strains are both MDR and sufficiently virulent, as we see on the example of strain 4542 / SIT266. Which is why we write: “More number of resistance mutations is in negative correlation with virulence in SIT252 but the situation was more complex with strain SIT266 which is both the most virulent and bears 10 drug resistance mutations.”

-Fig. 2A, correct colour code (red line in the legend for 4542 strain)

ANSWER: corrected.

-p.i. upon first use, specify abbreviation

ANSWER: done (section *Animal study design* in M&M).

August 6, 2023

Dr. Igor Mokrousov
St. Petersburg Pasteur Institute
Laboratory of Molecular Epidemiology and Evolutionary Genetics
14 Mira street
St. Petersburg 197101
Russia

Re: Spectrum01392-23R1 (A multifaceted interplay between virulence, drug resistance and phylogeographic landscape of *Mycobacterium tuberculosis*)

Dear Dr. Igor Mokrousov:

Your manuscript has been accepted, and I am forwarding it to the ASM Journals Department for publication. You will be notified when your proofs are ready to be viewed.

Sincerely,

Alexandra Aubry
Editor, Microbiology Spectrum
